# The Transcription Factor EB Reduces the Intraneuronal Accumulation of the Beta-Secretase-Derived APP Fragment C99 in Cellular and Mouse Alzheimer’s Disease Models

**DOI:** 10.3390/cells9051204

**Published:** 2020-05-12

**Authors:** Anaïs Bécot, Raphaëlle Pardossi-Piquard, Alexandre Bourgeois, Eric Duplan, Qingli Xiao, Abhinav Diwan, Jin-Moo Lee, Inger Lauritzen, Frédéric Checler

**Affiliations:** 1IPMC UMR 7275 CNRS/UCA, Laboratory of Excellence DistALZ, 660 route des Lucioles, 06650 Valbonne, France; becot@ipmc.cnrs.fr (A.B.); pardossi@ipmc.cnrs.fr (R.P.-P.); bourgeois@ipmc.cnrs.fr (A.B.); duplan@ipmc.cnrs.fr (E.D.); lauritzen@ipmc.cnrs.fr (I.L.); 2Department of Neurology and the Hope Center for Neurological Disorders, Washington University School of Medicine, St. Louis, MO 63110, USA; xiaoq@neuro.wustl.edu (Q.X.); leejm@neuro.wustl.edu (J.-M.L.); 3Department of Medicine, Washington University School of Medicine, St. Louis, MO 63110, USA; adiwan@dom.wustl.edu; 4John Cochran Veterans Affairs Medical Center, St. Louis, MO 63106, USA

**Keywords:** Alzheimer’s disease, C99, βCTF, TFEB, 3xTgAD mice, AAV8, cathepsins, lysosomes

## Abstract

Brains that are affected by Alzheimer’s disease (AD) are characterized by the overload of extracellular amyloid β (Aβ) peptides, but recent data from cellular and animal models propose that Aβ deposition is preceded by intraneuronal accumulation of the direct precursor of Aβ, C99. These studies indicate that C99 accumulation firstly occurs within endosomal and lysosomal compartments and that it contributes to early-stage AD-related endosomal-lysosomal-autophagic defects. Our previous work also suggests that C99 accumulation itself could be a consequence of defective lysosomal-autophagic degradation. Thus, in the present study, we analyzed the influence of the overexpression of the transcription factor EB (TFEB), a master regulator of autophagy and lysosome biogenesis, on C99 accumulation occurring in both AD cellular models and in the triple-transgenic mouse model (3xTgAD). In the in vivo experiments, TFEB overexpression was induced via adeno-associated viruses (AAVs), which were injected either into the cerebral ventricles of newborn mice or administrated at later stages (3 months of age) by stereotaxic injection into the subiculum. In both cells and the 3xTgAD mouse model, exogenous TFEB strongly reduced C99 load and concomitantly increased the levels of many lysosomal and autophagic proteins, including cathepsins, key proteases involved in C99 degradation. Our data indicate that TFEB activation is a relevant strategy to prevent the accumulation of this early neurotoxic catabolite.

## 1. Introduction

Alzheimer’s disease (AD) is the most common form of neurodegenerative dementia and has been mainly associated with the accumulation of extracellular plaques composed of aggregated Aβ and intracellular neurofibrillary tangles (NFTs) composed of hyperphosphorylated tau. Aβ is derived from the amyloid precursor protein (APP) by sequential proteolytic processing by β- and γ-secretases [1]. The initial cleavage of APP by β-secretase results in the secretion of the soluble ectodomain sAPPβ and the generation of a membrane-tethered C-terminal fragment C99 (or βCTF) that is cleaved by γ-secretase to form Aβ. Increasing evidence indicates that Aβ accumulation is preceded by intraneuronal accumulation of C99, which contributes to AD pathology by affecting the endolysosomal network [2,3,4,5,6,7,8,9,10,11]. Firstly, in brain fibroblasts derived from Down’s syndrome (DS) patients, a disorder in which patients develop an early form of AD likely due to a third copy of APP present on chromosome 21 [12], C99 accumulation was found to cause enlargement of Rab5-positive early endosomes [2,3,4]. Strikingly, these alterations resembled those described years before in post-mortem brains from early-stage AD-affected patients [13]. Later on, it was shown that C99 accumulation also induces lysosomal-autophagic pathology in both cellular and animal AD models [5,6,8,10]. Importantly, both of these C99-induced defects are also observed in human iPSC (induced pluripotent stem cell) neurons carrying endogenous levels of familial *APP* or *PSEN1* mutations, indicating that these alterations occur in the absence of overexpressed APP [8,9]. For instance, late endosomal and lysosomal pathologies were observed in monogenic iPSCs derived from AD patients carrying *APP* or *PSEN1* mutations [8] and early endosome defects were demonstrated in isogenic *APP* and *PSEN1* mutant iPSCs generated by CRISPR/Cas9 knock-in [9]. Interestingly, in all these reports, alterations were mediated by C99 itself and not by Aβ, since the pharmacological inhibition of the β-secretase BACE1 (that prevents formation of both C99 and thereby Aβ) abrogated these defects, whereas they were amplified by the blockade of γ-secretase (that halts Aβ generation and thus enhances C99 levels). Moreover, recent studies showed that these C99-induced endosomal alterations are associated with aberrant neurotrophic signaling and cholinergic neurodegeneration [7], brain network alterations, and neuronal hyperactivity [14], as well as with long-term potentiation deficits and cognitive impairment [6,15,16]. In addition, a recent study using brain tissue sections from sporadic AD-affected brains showed both an anatomical and quantitative correlation of C99, but not of Aβ, with the degree of neuronal vulnerability to neurodegeneration and cognitive impairment [17]. Overall, these reports consistently document a key role of C99 accumulation in AD pathogenesis and thus suggest that the inhibition of early AD-associated C99 accumulation should be beneficial [18].

Our recent studies suggest that C99 accumulation in the 3xTgAD mouse model is linked to a defective lysosomal degradation rather than to a lowered γ-secretase-mediated processing [6]. Thus, 2xTgAD mice (expressing wild-type presenilin) display similar C99 accumulation than 3xTgAD mice, although the levels of Aβ are barely detectable in the former model, indicating that C99 accumulation could not be directly due to reduced γ-secretase activity. Indeed, C99, as other APP-CTFs are known to be degraded by cathepsins through the lysosomal-autophagic degradation pathway [6,19,20,21], but lysosomal degradation is defective in AD [22]. The basic helix-loop-helix leucine zipper transcription factor EB (TFEB) is a master regulator of lysosome and autophagy biogenesis [23]. Under physiological conditions, TFEB undergoes mTOR-mediated phosphorylation and resides in the cytosol. Under aberrant lysosomal storage conditions, TFEB is dephosphorylated and, consequently, is translocated into the nucleus where it activates its target genes via the CLEAR (Coordinated Lysosomal Expression and Regulation) consensus sequence. As opposed to autophagic activators, TFEB promotes cellular clearance by regulating several steps of lysosomal-autophagic degradation including lysosome biogenesis, autophagosome formation, and autophagosome-lysosome fusion [23]. Indeed, TFEB is known to upregulate more than 290 genes including cathepsins, the main proteases involved in lysosomal APP-CTF degradation [6,20]. A still increasing amount of literature shows that both exogenous expression of TFEB and pharmacological activation of endogenous TFEB can promote the selective clearance of intracellular neurotoxic proteins and can have beneficial effects in these diseases (for review see [24,25]). Thus, we aimed at investigating the potential of TFEB overexpression to lower C99 expression in AD cellular models as well as a mean to interfere with early intraneuronal accumulation in vivo, in the 3xTgAD mouse model.

## 2. Materials and Methods

### 2.1. Animals and Viral Infection

3xTgAD mice (harboring APP_swe_, PS1_M146V_, and Tau_P301L_ transgenes) [26] were generated from breeding pairs provided by Dr. La Ferla (Irvine, CA, USA). AAV8 viral particles (generated by the Hope Center Viral Core at Washington University) driving expression of TFEB (AAV8-cmv-FLAG-TFEB) or GFP (AAV8-cmv-GFP) (2.8 × 10^12^ vg/mL (viral genomes per mL)) [27] were delivered in vivo by means of two distinct protocols. In the first protocol, AAVs were injected into the ventricles of newborn mouse brains as described previously [6,28]. Briefly, newborn (post-natal day 0, P0) females were injected unilaterally into ventricles with 2.5 µL of AAV8-cmv-FLAG-TFEB (n > 30) or AAV8-cmv-GFP (n > 30) and mice were analyzed at 8 months post-AAV delivery. In the second protocol, AAVs were delivered by stereotaxic injection into the subiculum of 3-month-old females (n = 8). Mice were anesthetized by intraperitoneal injection of ketamine (100 mg/kg) and xylazine (24 mg/kg) and placed in a stereotaxic apparatus. Viruses (2 µL per site) were then injected bilaterally into the subiculum at the following coordinates from bregma AP, −3.8, ML, 2.5, and DV, 1.5 at a flowrate of 0,4 µL/min. Animals were euthanized 5 months post-injection, e.g., at 8 months of age. For immunohistochemistry, mice were anesthetized as described above and intracardiacally perfused with PBS followed by 4% paraformaldehyde before collecting the brains. All animals were housed with a 12:12 h light/dark cycle and were given free access to food and water and experimental procedures were in accordance with the European Communities Council Directive of 24 November 1986 (86/609/EEC) and approved by Nice University Animal care and use Committee, and the National Council on Animal care of the Ministry of Health (Project Number: APAFiS#17993-2018112613266973 v2).

### 2.2. Cell Culture and Treatments

Human neuroblastoma overexpressing Swedish mutated APP (SH-SY5Y-APPswe) [29], or human embryonic kidney cells (HEK293, ATCC), naïve or stably overexpressing APPswe (HEK-APPswe) [30] or C99 (HEK-C99) [31]), were cultured in Dulbecco’s modified Eagle’s medium supplemented with 10% fetal calf serum, penicillin (100 U/mL), and streptomycin (50 µg/mL) purchased from Life Technologies (Villebon sur Yvette, France) and with G418 or puromycin, respectively, for stable lines at 37 °C/5% CO_2_. One day before transfection, cells were either plated into 6-well plates (biochemical studies) or were plated into poly-D-lysine coated coverslips placed in 12-well plates (for immunocytochemistry). Transient transfections of cells were carried out using lipofectamine 2000 (Life Technologies) for SH-SY5Y and JetPrime (Polyplus transfection, Ozyme, Montigny le Bretonneux, France) for HEK293 cells, according to the manufacturer’s instructions, and cells were recovered 48 h post-transfection. SH-SY5Y-APPswe and HEK-APPswe were transfected with FLAG-tagged murine TFEB cDNA or the empty pAAV vector containing a CMV promoter [27], whereas naïve HEK cells were co-transfected with C99 and TFEB cDNA (or empty vector).

### 2.3. Immunolabeling of Tissue and Cultured Cells

Paraformaldehyde-fixed brains (see above) were embedded in paraffin and cut on a microtome in 8 µm thick sections (Thermoscientific, Illkirch, France). Brain sections were treated with formic acid for 5 min followed by heat treatment in citrate buffer during 30 min. After saturation in 5% BSA/0.1% Triton, slices were incubated at 4 °C overnight with primary antibodies (α-TFEB (rabbit polyclonal, Bethyl 1:1000), α-CatB (rabbit monoclonal, Abcam 1:1000), α-CatD (rabbit monoclonal, Abcam 1:1000), α-NeuN (rabbit monoclonal, Abcam 1:2000), α-Tuj1 (mouse monoclonal, BioLegend, 1:1000), 82E1 (mouse monoclonal, IBL 1:200), α-APPct (rabbit polyclonal, gift from Dr Fraser, 1:1000)) followed by HRP-conjugated antibodies (Jackson ImmunoResearch, 1:1000) and DAB substrate (DAB impact, Vector LAbs), Alexa Fluor-488 or Alexa Fluor-647 conjugated anti-mouse or anti-rabbit antibodies (Molecular Probes, 1:1000) and DAPI (Roche, 1:20 000). Other slices were used for the 4-color IHC Opal Multiplex assay (Perkin Elmer, Villebon sur Yvette, France) allowing highly sensitive protein detection using primary antibodies from the same species. Slices were pretreated as described above, incubated sequentially with primary antibodies for 1 h at room temperature and secondary antibodies from the OPAL kit (1/1000) and stripped at 95 °C in citrate buffer for 20 min. When using the OPAL assay, it was possible to label TFEB, cathepsin B, or cathepsin D as well as C-termini of APPct on the same slices. Cells in culture were fixed with paraformaldehyde, permeabilized with 0.1% Triton-X 100 for 10 min, saturated in 5% BSA/0.1% Triton-X 100, probed 1 h with primary antibodies (for TFEB detection with α-TFEB (Bethyl 1:5000) or α-Flag (Sigma 1:5000) and for APP-CTF detection with α-APPct, (1:5000), 6E10 (Covance, 1:1000) or 82E1 (IBL, 1:1000)) followed by Alexa Fluor-488 and Alexa Fluor-594 conjugated antibodies (Molecular Probes, 1:1000) and DAPI (1:20,000) staining. Confocal images were acquired using a Zeiss SP5 confocal (Marly le Roi, France) or an Olympus Fluoview10 microscope (Rungis, France) for cells and an Olympus Fluoview10 confocal microscope for tissue sections. Images of tissue sections revealed with the DAB substrate were photographed using a bright field microscope (DMD108, Leica, Paris, France,).

### 2.4. Immunofluorescence Quantification

SH-SY5Y-APPswe cells were transfected with FLAG-tagged murine TFEB cDNA or the empty pAAV vector containing a CMV promoter [27] and immunolabeled at 48 h post-transfection with α-Flag (for TFEB detection) and α-APPct (for APP-CTF detection). Images were acquired randomly using an Olympus Fluoview10 microscope (Rungis, France) and an objective 40×. A total of 6–7 images (about 100 cells each) were acquired for each condition (in 3 independent experiments). Quantification of the intensity of APP-CTF immunolabeling was performed using a home-made quantification program allowing the detection of the total number of cells (visualized with DAPI nuclei, blue staining), TFEB-positive cells (green labeling), and APPct immunoreactivity (red staining). The quantification of APPct staining intensity was compared in TFEB-positive (green) versus TFEB-negative (blue but not green) cells. In the in vivo experiments, 82E1 immunostaining was estimated by a visual inspection of TFEB-positive neurons (in stereotaxic-injected mice, n = 250 cells) and staining was determined as being absent, low or normal (as compared to that of the surrounding TFEB-negative neurons).

### 2.5. Western Blot Analysis

Cell homogenates (20–40 µg) were separated by Bio-Rad 12% stain-free^TM^ TGX FastCast^TM^ acrylamide gels or 16% tris-tricine gels (for the detection of APP-CTFs). Bio-Rad gels were photoactivated for the visualization of proteins before being electrophoretically transferred to nitrocellulose membranes using the Bio-Rad Trans-Blot^®^ Turbo^TM^ Transfer System. Tris-tricine gels were directly transferred to nitrocellulose membranes using a conventional transfer system and boiled in PBS before saturation with skimmed milk. Membranes were blotted with the following antibodies: α-APPct (1:1000), α-FLAG (1:5000), α-TFEB (1:1000), α-CatD (1:1000), α-LC3 (Novus. 1:1000), or α-Actin (Sigma, 1:5000). After probing with primary antibodies, immunological complexes were revealed with HRP-conjugated antibodies (Jackson ImmunoResearch, 1:10,000) followed by electrochemiluminescence (Westernbright^TM^ Sirius^TM^ and Quantum^TM^ chemiluminescent HRP substrate, Advansta, France). Peak height of signal intensities from protein bands were quantified with ImageJ software.

### 2.6. mRNA Extraction and Quantitative RT-PCR

Hippocampal tissue was lysed using magnetic beads and the MagNA lyser (Roche, Meylan, France). RNA from hippocampal tissue or cells was isolated using RNeasy Plus-Universal kit and RNeasy mini kit (Qiagen, Les Ulis, France), respectively. cDNAs were synthesized from 2 µg total RNA using AMV reverse transcriptase (Promega, GoScript) and oligo-dT primers. Then, samples were subjected to real-time PCR by means of a Rotor-Gene 6000 apparatus (Qiagen, Les Ulis, France), using the SYBR Green Roche detection protocol. Gene-specific primers (Table 1) were designed with the Universal Probe Library Assay Design Center software (Roche Applied Science, France). Relative expression analysis was normalized against two reference genes, the ribosomal gene RPL19 and glyceraldehyde-3-phosphate dehydrogenase (GAPDH) (see Table 1), which were amplified in parallel.

### 2.7. Statistics

All quantitative data were subjected to non-parametric tests, the Mann–Whitney U test for single comparison using GraphPad Prism 7. Data are represented as means ± SEM. Statistical significances were set up as * *p* < 0.05, ** *p*< 0.01 and *** *p* < 0.001.

## 3. Results

### 3.1. TFEB Overexpression Reduces APP-CTF Levels in both APPswe and C99 Expressing Cells

In order to determine the effects of TFEB overexpression on APP-CTF levels, we first transfected SH-SY5Y-APPswe cells with TFEB cDNA or an empty vector (Figure 1). Immunoblot analysis revealed that TFEB protein expression was higher in cell homogenates at 24 h than at 48 h post-transfection, (Figure 1A), while TFEB-mediated transcriptional effects peaked 48 h after transfection, as observed for cathepsin B (Figure 1A), suggesting a time-dependent shifting between TFEB protein expression and transcriptional function. Accordingly, quantitative PCR aimed at analyzing the effects of TFEB overexpression on TFEB-induced lysosomal and autophagic genes [23] was performed at 48 h post-transfection. In SH-SY5Y-APPswe neuroblastoma, TFEB induced an upregulation of several lysosomal and autophagic mRNAs (Figure 1B), but interestingly, it had strong effects on the expression of cathepsin mRNAs and particularly on that of CTSD encoding cathepsin D, one of the main proteases involved in C99 degradation. As previously described, TFEB also enhanced the transcriptional expression of the lysosomal membrane proteins LAMP1 and LAMP2B and of the autophagic genes MAP1LC3B (LC3) and SQSTM1 (p62) (Figure 1B). TFEB overexpression did not affect APP mRNA expression (Figure 1B). These findings were in agreement with immunocytochemical analysis revealing that overexpression of TFEB (detected by either α-FLAG (Figure 1C) or α-TFEB (Figure 1F)) was sufficient to activate it, e.g., to drive it into the nucleus to activate the transcription of cognate genes, as we have previously described [27]. Next, to assess the influence of TFEB expression on APP and APP-CTFs, we performed co-immunostaining using α-FLAG and α-APPct (Figure 1C). Strikingly, cells devoid of TFEB displayed strong α-APPct immunoreactivity whereas this immunostaining was barely detectable in TFEB-positive cells (Figure 1C). Indeed, the quantification of α-APPct immunoreactivity revealed about 80% reduction in TFEB-positive cells as compared to TFEB-negative cells (Figure 1D). This TFEB-mediated effect was also observed when immunocytochemical analysis was performed by means of the N-terminal directed antibodies 82E1 and 6E10 (Figure 1D). To more precisely quantify these effects on C99 levels, we switched into HEK293 cells to obtain higher gene transfection efficiency and performed qPCR and immunoblot analysis (Figure 2). Indeed, in HEK293 cells TFEB gene expression was 10 times higher than in SH-SY5Y cells. qPCR analysis indicated that TFEB triggered the induction of lysosomal and autophagic gene expressions mostly similarly in HEK293 and SY-SY5Y cells (Figure 2A) with the noticeable exception of CSTB mRNA levels that, unlike in SH-SY5Y cells, remained unaffected in HEK293 cells. Thus, although comparison of TFEB-mediated phenotypes unraveled some cell-specific transcriptional modulations, it was important to note that the APP-CTF-degrading CSTD mRNA was strongly upregulated in both cell lines, indicating that HEK293 cells were a relevant model to study TFEB-mediated regulation of APP-CTF expressions. In this context, we combined three complementary cell models: first, a monoclonal HEK293 cell-line stably expressing APPswe [30] and second, an inducible polyclonal HEK293 cell-line stably expressing C99 [31], two cell-lines which were both transfected with TFEB or empty vector (Figure 2B–E). Third, naïve HEK293 cells were co-transfected with C99 and TFEB cDNA (or empty vector) (Figure 2F,G). Whatever the cellular model examined, TFEB overexpression consistently led to increased levels of cathepsin D and LC3 (Figure 2B,D,F) and more interestingly, TFEB overexpression also triggered a significant reduction in both C99 and C83 levels in all models in accordance with the immunocytochemical analysis (Figure 2B–G). Moreover, while TFEB had little if any effect on the levels of immature APP, it also significantly reduced those of both overexpressed (Figure 2B,C) and endogenous (Figure 2F,G) mature APP, as we previously observed [27].

### 3.2. TFEB Overexpression Lowers C99 Accumulation in the Triple-Transgenic Mouse Model

Our previous works have proposed that C99 accumulation in the 3xTgAD is a consequence of defective lysosomal-autophagic degradation rather than a defective γ-secretase processing [6]. Although these data revealed increased levels of both autophagic and lysosomal proteins including cathepsins, in the 3xTgAD mouse, as compared to non-transgenic mice, we found a decrease in lysosomal function when estimated by cathepsin (B and D) enzymatic dosages. Here, in order to evaluate the functional status of TFEB in these mice, we performed qPCR measurements of some of the TFEB regulated genes (Appendix A). This analysis revealed a significant upregulation of TFEB itself, as well as of CSTB and CSTD mRNAs, in 3xTgAD mouse hippocampi as compared to age-matched controls. These data propose that TFEB signaling is upregulated, rather than downregulated in these mice. Thus, we investigated the potential of TFEB overexpression to reduce this C99 accumulation. For that purpose, we used two distinct strategies based on viral-mediated delivery of TFEB by means of AAV8-cmv-FLAG-TFEB [27], which were administrated into mice at different stages: at birth or at 3 months of age, the latter corresponding to an age at which C99 accumulation has started. In the first protocol, AAV8-cmv-FLAG-TFEB or control AAV8-cmv-GFP were delivered into the cerebral ventricles of neonatal 3xTgAD mouse brains. This technique is described to ensure widespread and long-lasting protein expression throughout the whole brain [32] and is routinely used in our own laboratory for injecting AAV-10 particles [6]. Previous work has shown a critical role of the timing of AAV injection, since it will determine both the biodistribution and tropism of the gene product [32]. Indeed, it was observed that when injected in newborns older than about 24–48 h, AAV8 particles have a lower degree of diffusion and a preference for glial transduction [32]. Indeed, we confirmed these observations and found that AAV8-cmv-GFP injected into mice older than 24 h (P1) mostly exhibit glial GFP expression, whereas the same viruses injected in younger mice (less than 24 h after birth, P0), display almost exclusive neuronal expression, as visualized by morphology inspection of 3-week-old mice (Appendix A). In these mice GFP expression was observed throughout the brain including in the subiculum, the area displaying C99 accumulation in the 3xTgAD mouse model [11]. Thus, in our experiments AAV8-cmv-FLAG-TFEB or control AAV8-cmv-GFP were injected only at P0 in order to ensure neuronal expression. Then mice were analyzed by immunohistochemistry at 8 months post-AAV-delivery, an age at which intraneuronal staining with 82E1 still corresponds mostly to C99 [11]. Noticeably, while all animals displayed high GFP expression within the subiculum, a much fewer number of animals displayed significant TFEB expression within this area, although a similar number of viral particles were injected. Although in most animals, TFEB expression was low in the subiculum, its expression was high in other brain areas, suggesting a lower diffusion capacity, rather than a lower stability, of the TFEB viruses as compared to the GFP viruses (not shown). Moreover, and in agreement with this hypothesis, the same differences were observed when animals were analyzed at much earlier timepoints. However, as observed for GFP, in these “TFEB-positive” animals, nearly all TFEB-positive cells were neurons as observed by colocalization with the neuronal marker NeuN (Figure 3A). Moreover, high magnification images indicated that many neurons display nuclear TFEB as observed with DAPI staining (Figure 3A, right panels). Thus, we selected the cohort of animals displaying significant TFEB expression within the subiculum to follow lysosomal activation and monitor C99 accumulation. C99 detection on paraffin sections requires formic acid treatment [11] and we noticed that this treatment led to a complete depletion of GFP-associated fluorescence in AAV-cmv-GFP animals probably due to formic acid induced membrane permeabilization. This observation, however, allowed us to use green-fluorescence conjugated antibodies for the detection of other proteins. To determine the effect of TFEB on lysosomal activation, we performed immunohistochemistry with α-cathepsin B antibodies on slices adjacent to those labeled with α-TFEB. In the animals expressing the highest levels of TFEB (n = 4), we observed a clear increase in cathepsin B staining (Figure 3B, middle panel) within the same areas positive for TFEB (Figure 3B, upper panel). Moreover, in these animals, co-immunostaining of TFEB and 82E1 revealed a strong reduction in 82E1-immunostaining, then indicating a decrease in C99 levels (Figure 3, lower panels). In all other analyzed animals (n > 20), the expression of TFEB was too low within the subiculum to induce both changes in cathepsin and C99 staining, indicating that these changes were dependent on a high level of TFEB overexpression.

In the second protocol, AAV8s were injected stereotaxically and bilaterally into the subiculum of 3-month-old animals, an age at which C99 can be consistently detected [11] (Figure 4 and Figure 5). Here again, immunohistochemical analysis was performed at 8 months of age. First, this analysis confirmed a strong and widespread expression of TFEB in the subiculum. Unlike what was observed in icv-injected newborn mice, TFEB was expressed in all injected mice and within almost the entire subiculum (n = 8) (Figure 4A). Again, higher magnification images of the subiculum showed that TFEB overexpression enhanced its nuclear localization i.e., activation. Both the morphological shape of TFEB-positive cells and co-labeling with NeuN or Tuj1 (Figure 4B) indicated that most of these cells indeed corresponded to neurons. Here again, the comparison of AAVcmv-GFP and AAV-cmv-FLAG-TFEB-injected animals indicated that the overexpression of TFEB strongly reduced C99 accumulation, as visualized both by both HRP-conjugated antibodies and DAB development (Figure 5A) and after co-immunostaining of TFEB and C99 and protein visualization with fluorescent-conjugated antibodies (Figure 5B,C). In both cases, C99 staining was clearly reduced in TFEB-positive mice. High magnification images of co-immunostaining of TFEB and 82E1 indicated that only TFEB-positive neurons displayed reduced 82E1 immunoreactivity (see arrows in Figure 5C), whereas in the same areas, TFEB-negative neurons still displayed 82E1 staining. To obtain a more quantitative estimation of this TFEB-mediated effect, we evaluated the 82E1 staining in TFEB positive cells, and found a total absence of staining in nearly 90% of TFEB-positive cells and a reduced staining in about 10% of the remaining TFEB-positive cells, as compared to the staining in the surrounding neurons. A strongly reduced immunostaining in TFEB-positive neurons were also seen when using α-APPct (Figure 5C).

## 4. Discussion

Lysosomal-autophagic dysfunction is known to be a frequent feature of AD, which precedes the classical late-stage hallmarks Aβ plaques and neurofibrillary tangles [22]. The transcription factor EB (TFEB) is one of the key regulators of lysosomal-autophagic function [23]. TFEB regulates both lysosomal biogenesis and autophagy by coordinating the expression of lysosomal hydrolases, lysosomal membrane proteins, as well as autophagy-related genes in responses to lysosomal stress conditions. Numerous studies suggest that the lysosomal-autophagic dysfunction in AD is related to a defective TFEB function and, accordingly, propose TFEB activation as a possible strategy to counteract disease progression in this pathology. However, contrasting findings regarding the TFEB status in AD brain tissue have been reported. One study reported that TFEB is retained within the cytoplasm and lowered in nuclear fractions of hippocampi from late-stage Braak AD brains [33], leading to reduced TFEB-mediated activation of the CLEAR network. A similar situation was described in Parkinson’s disease, in which α-synuclein acts as a sequestering molecule and impedes the translocation of TFEB to the nucleus [34]. There are no evidences of similar effects on TFEB sequestering by aggregate-prone proteins in AD, but other mechanisms leading to TFEB dysfunction have been proposed. One study demonstrated that TFEB function is dependent on presenilin 1 and is negatively impacted by AD-associated *PSEN1* mutations, a mechanism which may explain TFEB dysfunction at least in some familial AD cases [35]. Another study showed that apolipoprotein E4, the main AD susceptibility risk factor [36], binds to the CLEAR motif and competes with TFEB binding, thereby suppressing autophagy activation in apolipoprotein (APOE) e4/e4 carriers [37]. In contrast to these studies, others have reported a gain of TFEB activity in AD. For instance, microarray analysis of CA1 neurons from AD patients revealed an up-regulation of a high number of autophagic genes regulated by TFEB [38]. Moreover, immunohistochemistry showed a higher number of cells displaying nuclear TFEB [38] in pathological brains. Similar up-regulation of autophagic genes was observed in transcriptomic analysis of brain of the 5xFAD mouse model [39]. These apparently discrepant observations could well be reconciled if one envisions a bell-shaped expression and/or function of TFEB in pathological conditions. Thus, early activation of TFEB could be seen as an initial neuroprotective response to cell stress aimed at circumscribing early accumulation of aggregated proteins, but which ultimately becomes functionally disrupted or not sufficient to counteract protein accumulation. In the present study, we observed that in the 3xTgAD mouse model many neurons displayed nuclear TFEB (at least when overexpressed, since we were unable to detect endogenous TFEB, in agreement with a previous paper indicating low endogenous TFEB levels in neurons [38]). Moreover, when we compared the endogenous gene expression in age-matched non-transgenic and 3xTgAD hippocampi by qPCR measurements, we found an increase in many TFEB targeted genes including CSTB, CSTD and LAMP1, as well as in TFEB itself (suppl. Figure 1A) in the 3xTgAD model, which would then be in favor with upregulated rather than downregulated TFEB signaling, although we cannot exclude the involvement of the other members of the Mit/TFE3 family in this regulation [38]. Nevertheless, our previous works have indicated defects in cathepsin function and substrate degradation, suggesting that, as hypothesized above, this up-regulation of TFEB signaling could be seen as a compensatory mechanism, which however in the 3xTgAD mouse is not sufficient to circumscribe protein accumulation and prevent a pathological process [6].

Although it remains unclear to what extend TFEB function is defective in AD, previous studies from AD cell and mouse models indicate that both the pharmacological and genetic activation of TFEB could have beneficial effects. The first beneficial effect of TFEB overexpression on Aβ-related pathology was observed after TFEB-mediated stimulation of astrocytic Aβ degradation in the APP/PS1 mouse [40]. Another work, using an in vitro model of brain slices derived from APP/PS1 mouse brains, demonstrated that microglial expression of TFEB facilitated fibrillar Aβ degradation, an effect that was found to be further enhanced by deacetylation of TFEB by Sirt1 [41]. Moreover, neuronal overexpression of TFEB was found to reduce Aβ levels, an effect which was proposed to be due to an enhanced lysosomal degradation of endocytosed APP [27]. In the present work, we used AAV8-cmv-TFEB particles [27] to investigate the efficiency of TFEB in reducing C99 accumulation in the 3xTgAD mouse model, a model displaying early and consistent C99 accumulation in the absence of detectable Aβ [11]. Our previous studies suggested that this accumulation of C99 was a cause of a defective lysosomal degradation rather than a consequence of increased BACE1 activity or reduced γ-secretase processing [6]. Indeed, we here found that the overexpression of TFEB leads to decreased C99 levels in both cellular AD models as well as in the 3xTgAD mouse model. The reduced C99 accumulation was associated with an increased expression of several lysosomal and autophagic proteins, including cathepsin B and more importantly cathepsin D, the main proteases responsible for C99 degradation [6,20]. The lowered levels of C99 could not be due to a higher γ-secretase proteolysis of C99, since our previous work showed that TFEB had no effect on the expression of any of the members of the γ-secretase complex (presenilins, Nicatrin, Aph1, and Pen2) [27]. In cellular models expressing directly the C99 fragment in a wild-type genetic background, TFEB also decreased the levels of this fragment, indicating that this effect on C99 was due to a direct activation of lysosomal degradation of C99 and not only due to an enhanced degradation of endocytosed APP. In the 3xTgAD mouse model, a decrease of C99 was observed both when AAVs were injected into the cerebral ventricle of newborn mouse brains and when injected stereotaxically into the subiculum of 3-month-old mice. Taken together, these findings are important considering the numerous recent works suggesting that C99, independently on Aβ, is a key contributor to early AD pathology. Our data are in agreement with a very recent study demonstrating that targeting endogenous TFEB pharmacologically using the curcumin analogue (C1), leads to reduced APP, APP-CTFs, Aβ and tau aggregates in several mouse models including in the 3xTgAD mouse [42]. The effects of C1 indeed seemed to be directly linked to TFEB activation, since at least in cultured cells the knockdown of TFEB totally inhibited these beneficial effects. In agreement with our findings, in the 3xTgAD mouse, C1 treatment had a particularly strong effect on C99 levels and on the intraneuronal staining detected by 4G8. The work from Song and coll. also demonstrated that C1 treatment led to reduced cognitive defects in the 3xTgAD mouse. Thus, altogether our findings confirm that C99 levels are critically controlled by the lysosomal status and suggest that TFEB is a promising target to prevent the accumulation of this early neurotoxic catabolite.

## Figures and Tables

**Figure 1 cells-09-01204-f001:**
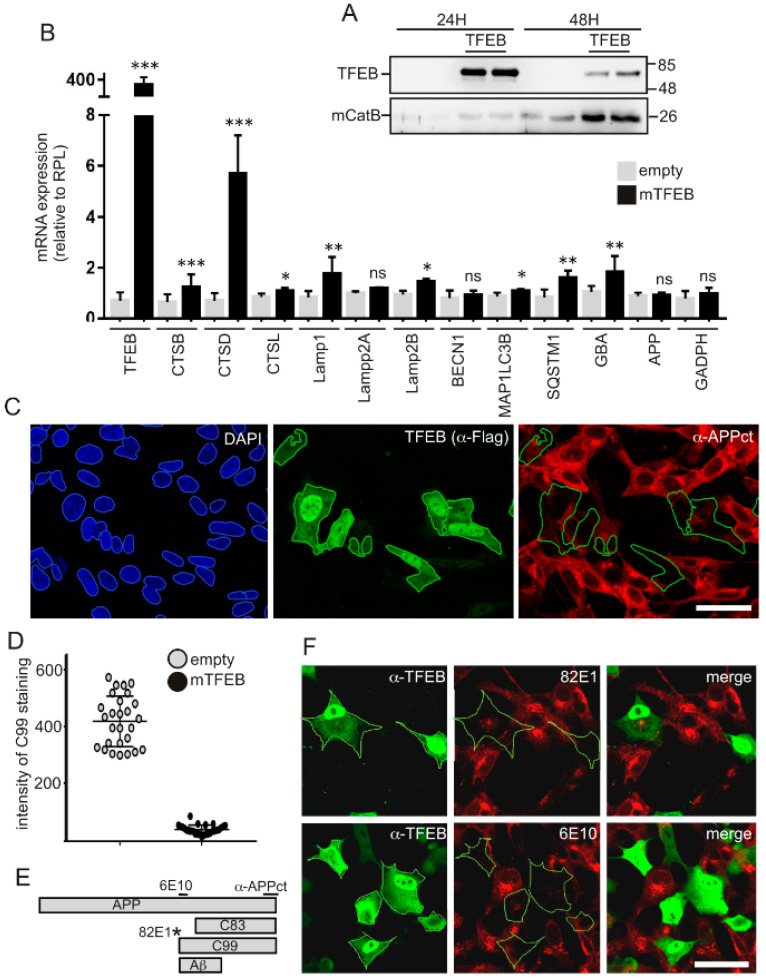
Transcription factor EB (TFEB) overexpression in SH-SY5Y-APPswe cells leads to induction of lysosomal-autophagic gene products and to reduced C99 levels. Statistical differences were * *p* < 0.05, ** *p* < 0.01, *** *p* < 0.001, ns = non significant (*p* > 0.05), versus control cells (empty) according to the Mann–Whitney test. (**A**) Immunoblot analysis revealed the levels of overexpressed TFEB and endogenous cathepsin B at 24 h or 48 h post-transfection. TFEB is detected using α-FLAG. (**B**) qPCR analysis of TFEB-mediated gene transcription showed the induction of numerous lysosomal-autophagic genes. Gene expressions were normalized to the expression of the housekeeping gene RPL19 and represented in each case relatively to mock-transfected cells (**C**) SH-APPswe cells were transfected with TFEB cDNA and immunostained at 48 h post-transfection with α-FLAG (for TFEB detection) and α-APPct. Nuclei were stained with DAPI. (**D**) Quantification of α-APPct staining in TFEB-positive versus TFEB-negative cells. (**E**) Illustration of epitope recognition by the different antibodies. (**F**) SH-APPswe cells were immunostained with α-TFEB (green labeling) and either 82E1 or 6E10 (red labelings). Bar scale: 10 µm.

**Figure 2 cells-09-01204-f002:**
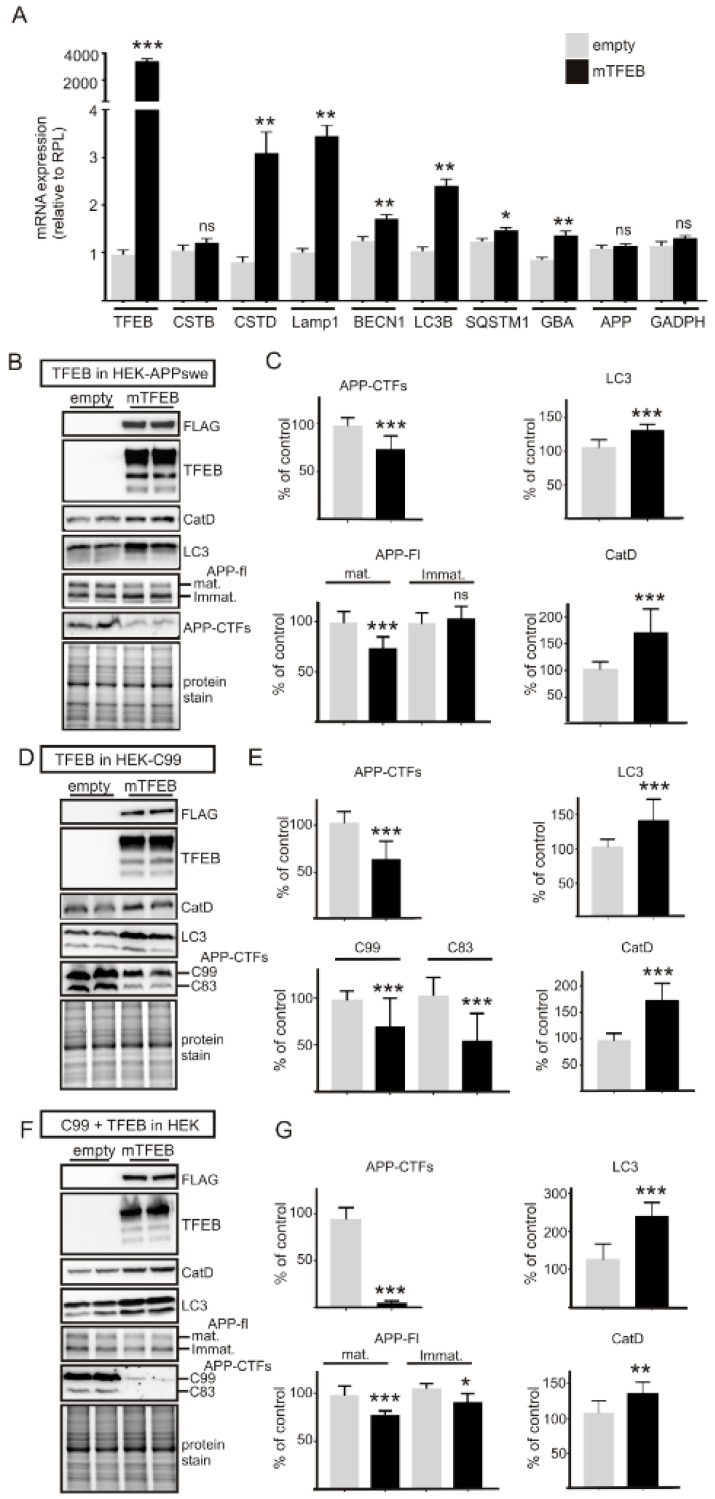
TFEB overexpression in HEK-293 APPswe or C99 expressing cells leads to the induction of numerous lysosomal-autophagic genes and to reduced C99 levels. (**A**) Bars show the quantification of qPCR analysis of TFEB-mediated gene transcription in naïve HEK-293-APPswe cells. (**B**–**G**) Immunoblot analysis of TFEB-mediated effects on lysosomal activation (cathepsin D and LC3 immunoreactivities) and APP/APP-CTF levels in HEK-293 cells stably expressing APPswe (**B**,**C**) or C99 (**D**,**E**), or in naïve HEK-293 cells transiently transfected with C99 and TFEB cDNA or empty vector (**F**,**G**). Bars in (**C**,**E**,**G**) correspond to the quantification of APP-CTFs, APPfl, LC3 and CatD (n = 12 from at least 3 independent determinations in each cell line) and are represented as means ± SEM and are expressed as the percent of control transfected cells (taken as 100). Statistical differences were * *p* < 0.05, ** *p* < 0.01, *** *p* < 0.001, ns = non significant (*p* > 0.05), versus control (empty) cells according to the Mann–Whitney test.

**Figure 3 cells-09-01204-f003:**
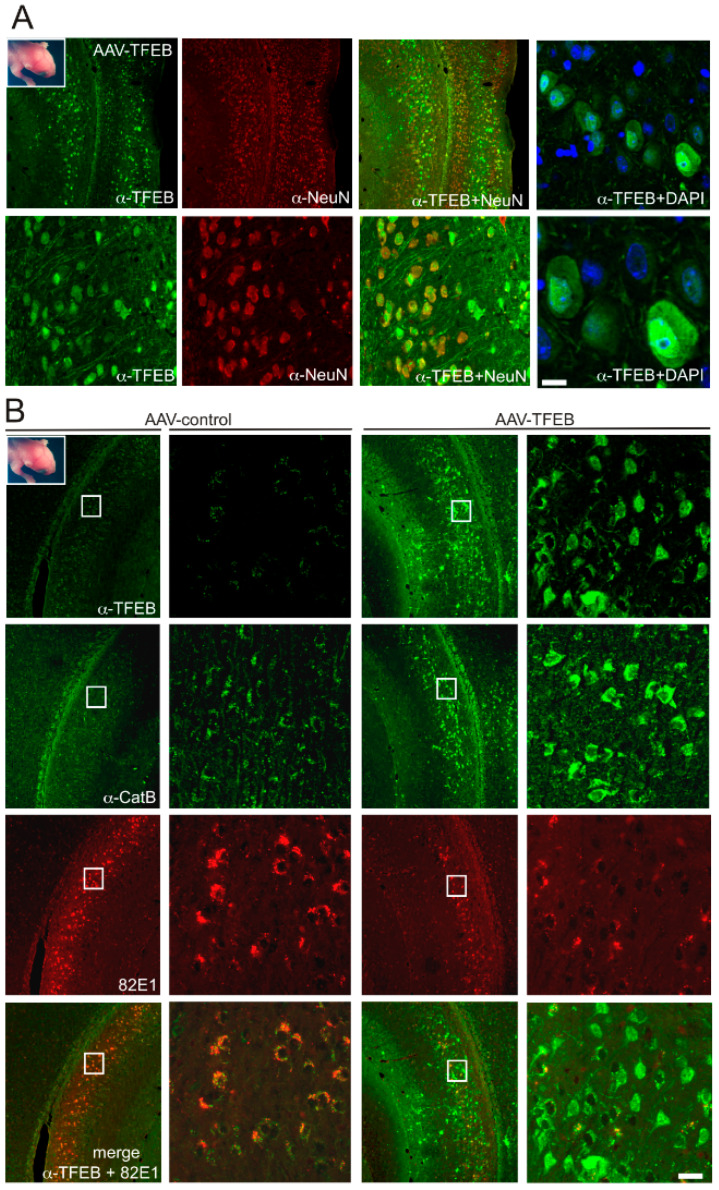
The overexpression of TFEB in the 3xTgAD mouse after intra-cerebro-ventricular injections of AAVs into neonatal 3xTgAD mice leads to neuronal and nuclear TFEB expression and is associated with increased cathepsin expression and a decrease in C99 accumulation. Mice were injected at P0 and analyzed by immunohistochemistry at 8 months of age. (**A**) Left panels show co-immunostaining of TFEB (detected with α-TFEB, green) and the neuronal marker NeuN (red). Right panels show nuclear localization of TFEB as colocalized with DAPI. (**B**) Images illustrate low and high magnification co-immunostaining of TFEB (α-TFEB, green) and C99 (82E1, red) in AAV-cmv-GFP (left panels) and AAV-cmv-FLAG-TFEB (right panels) mice. Lower panels show merged images of TFEB and 82E1. Middle panels show cathepsin B immunostaining, which was performed on adjacent brain slides. Bar scale: 100 µm, 20 µm, 5 µm and 2 µm in (**A**), and 100 µm and 10 µm in (**B**), respectively.

**Figure 4 cells-09-01204-f004:**
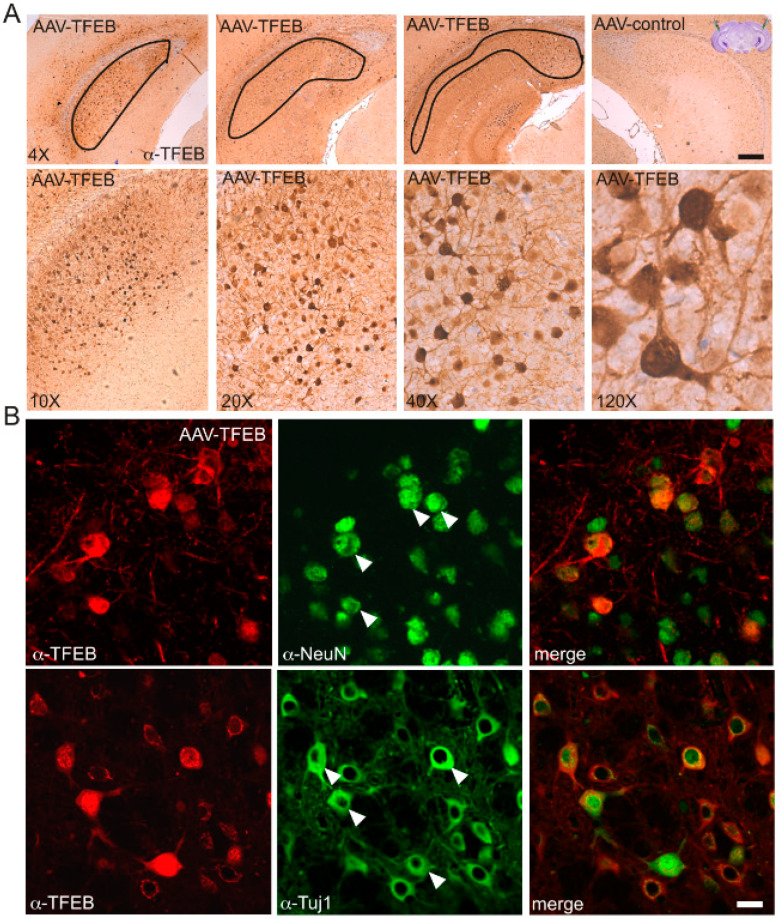
The overexpression of TFEB after stereotaxic injection of AAVs in the subiculum of 3-month-old mice leads to neuronal and nuclear TFEB staining. Immunostaining was performed in 8-month-old mice and revealed with HRP-conjugated antibodies and DAB (**A**) or with fluorescence-conjugated antibodies (**B**). (**A**) Images show the expression of TFEB at different levels of the subiculum in one of the AAV-cmv-FLAG-TFEB mice and the absence of staining in an AAV-cmv-GFP injected mouse (right panel). High magnification images show that the expression of TFEB is localized mainly in cells with the morphology typical of neurons and with a subcellular localization in both cytoplasm and nuclei. (**B**) Images show co-immunostaining of TFEB (red) and NeuN or Tuj1 (green) indicating that nearly all TFEB expression was localized to neurons. Scale bar: 150 µm (4×), 60 µm (10×), 30 µm (20×), 15 µm (40×) and 5 µm (120×) in (**A**) and 10 µm in (**B**).

**Figure 5 cells-09-01204-f005:**
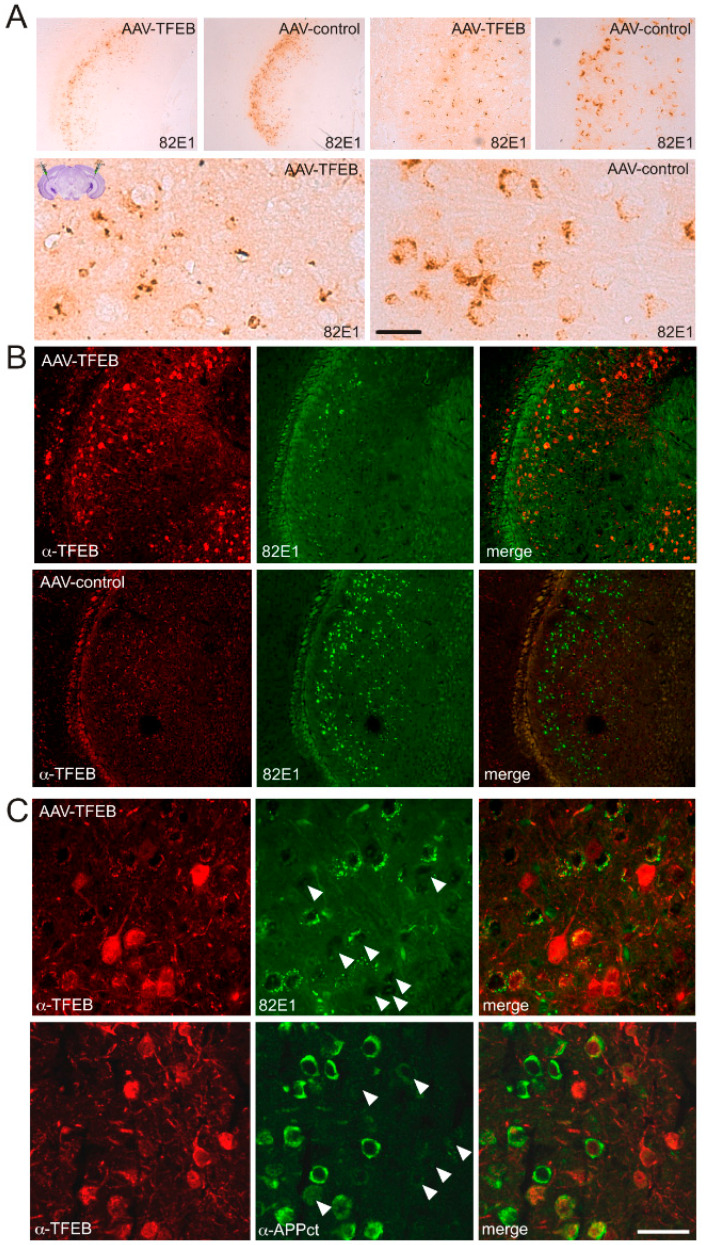
The overexpression of TFEB after stereotaxic injection of AAVs in the subiculum of 3-month-old 3xTgAD mice leads to decreased C99 levels. Mice were analyzed by immunostaining at 8 months of age. (**A**) Low- and high-magnification images illustrating the staining of 82E1 revealed with HRP-conjugated antibodies and DAB in an AAV-cmv-GFP or AAV-cmv-FLAG-TFEB injected mouse at the level of the subiculum. (**B**) Images of immunostaining of TFEB (red) and 82E1 (green) in 3xTgAD mice injected with either AAV-TFEB or AAV-GFP at the level of the subiculum. (**C**) High-magnification images revealing decreased 82E1 and α-APP-ct immunostaining within TFEB positive neurons (arrows). Scale bar: 500 µm), 100 µm and 25 µm in (**A**) and 100 µm and 20 µm in (**B**).

**Table 1 cells-09-01204-t001:** List of primers used for quantitative RT-PCR.

Gene name	Forward Primer	Reverse Primer
Human		
*TFEB*	5′-CCAGAAGCGAGAGCTCACAGAT-3′	5′-TGTGATTGTCTTTCTTCTGCCG-3′
*CSTB*	5′-TCATGGCCGAGATCTACAAA-3′	5′-TGACGTGTTGGTACACTCCTG-3′
*CSTD*	5′-GAAAGGCCTACTGGCAGGT-3′	5′TCCCTGTGTCCACAATAGCC-3′
*CSTL*	5′-TCCTATCCATATGAGGCAACAG-3′	5′-GTTGCAACTGCCTTCATCAG-3′
*LAMP1*	5′-ACGTTACAGCGTCCAGCTCAT-3′	5′-TCTTTGGAGCTCGCATTGG-3′
*LAMP2a*	5′-CAACAAAGAGCAGACTGTTTCAG-3′	5′-GCACTGCAGTCTTGAGCTGT-3′
*LAMP2b*	5′-AAGGGTTCAGCCTTTCAATG-3′	5′-CAACTATAATTGGGATTAGAATGGTGT-3
*MAP1LC3B*	5′-GAGAAGCAGCTTCCTGTTCTGG-3′	5′-GTGTCCGTTCACCAACAGGAAG-3′
*BECN1*	5′-GGCTGAGAGACTGGATCAGG-3′	5′-CTGCGTCTGGGCATAACG-3′
*SQSTM1*	5′-GCACCCCAATGTGATCTGC-3′	5′-CGCTACACAAGTCGTAGTCTGG-3′
*GBA*	5′-TCCTACCCTCGCCAACAGTA-3′	5′-GCTGCTTCTGGGTCTGTCA-3′
*APP*	5′-TTCATCATGGTGTGGTGGAG-3′	5′-GTTCTGCTGCATCTTGGACA-3′
*GAPDH*	5′-AGCCACATCGCTCAGACAC-3′	5′-GCCCAATACGACCAAATCC-3′
*RPL19*	5′GGGCATAGGTAAGCGGAAGG-3′	5′TCAGGTACAGGCTGTGATACA-3
Mouse		
*TFEB*	5′-GAGCTGGGAATGCTGATCC-3′	5′-GGGACTTCTGCAGGTCCTT-3′
*CSTB*	5′-CACACTGAAACCAGGCCTTT-3′	5′-CTTGCTGTGGTATCCAGTGTG-3′
*CSTD*	5′-GCCTACTGGCAGGTCCAC-3′	5′-GTGTCCACAATGGCCTGAC-3′
*LAMP1*	5′-GTGGCAACTTCAGCAAGGA-3′	5′-GTGGGCACAAGTGGTGGT-3′
*GAPDH*	5′-TGTCCGTCGTGGATCTGAC-3′	5′-CCTGCTTCACCACCTTCTTG-3′

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
