# Peer review of "The Transcription Factor EB Reduces the Intraneuronal Accumulation of the Beta-Secretase-Derived APP Fragment C99 in Cellular and Mouse Alzheimer’s Disease Models"

_cells, 2020, doi:10.3390/cells9051204_

Round 1

Reviewer 1 Report

The manuscript named “The transcription factor EB reduces the intraneuronal accumulation of the beta-secretase-derived APP fragment C99 in cellular and mouse AD models” deals with the up-to-date topic – Alzheimer's disease and relevant mouse models.

The manuscript is written and designed well. However, the manuscript has a few shortcomings. The figures are too small and are very hard to read. The font throughout the manuscript is not uniform, please, correct it. Line 198 contains two following dots, please, correct it. The manuscript contains few grammatical and stylistic errors.

This study is interesting and has an impact on the scientific community. After minor revisions, I would recommend this manuscript to be accepted to the Cells.

Reviewer 2 Report

In this manuscript, Bécot et al. analyzed the effect of an over-expression of TFEB, a master gene for lysosomal/autophagic function, on the intracellular accumulation of the C99 fragment of APP. The study was performed both in cell lines, and in vivo in neurons of the 3xTgAD transgenic mouse model of Alzheimer’s disease. The subject is of very high interest, and the study was well designed. I have, however, some concerns that are detailed below.

Major points

  1. In the Methods or in the Results section, it would be important to have an idea of the number of mice that were analyzed. Particularly, line 284 it would be important to know which proportion of mice present “sufficient” TFEB expression, and the criteria that were used to determine what a sufficient expression is. It could be very important if one envision the therapeutic potential of the author’s findings.

  1. It would also be very important to have a quantification of the effect of TFEB over-expression on APP-ct labeling in vivo, as was done in vitro. For instance, in Fig. 5C there are clearly some neurons showing strong TFEB expression that also contain strong APP-ct-IR.

Minor points

  1. Introduction line 65-66. Here and in the discussion, the authors indicate that APP-CTFs are principally degraded by Cathepsins. It is not clear to me, as I thought that the degradation of ß-CTF was mainly carried out by presenilins, leading to a cytoplasmic fragment that escapes Cathepsins, and to Aß. Could the authors clarify this point? Along this line, it could be interesting to know, either from the author’s own work or from the literature, if TFEB regulates presenilin expression.

  1. Results Line 277 the authors explain the lower expression of TFEB compared to GFP by a reduced diffusion of the AAV. Could it be due to a lower stability of the over-expressed TFEB? This alternative explanation could also be relevant with respect to a potential therapeutic perspective.

  1. Results Fig. 2C it seems to me that the reduction of APP-fl is in the same order of magnitude as the reduction of APP-CTF.

  1. Line 223 it seems to me that there is no over-expression of APP in Fig. 2D.

Round 2

Reviewer 2 Report

The authors properly addressed the reviewer's comments.